# CPAP Intervention as an Add-On Treatment to Lipid-Lowering Medication in Coronary Artery Disease Patients with Obstructive Sleep Apnea in the RICCADSA Trial

**DOI:** 10.3390/jcm11010273

**Published:** 2022-01-05

**Authors:** Yeliz Celik, Baran Balcan, Yüksel Peker

**Affiliations:** 1Research Center for Translational Medicine (KUTTAM), Koc University, 34010 Istanbul, Turkey; yuksel.peker@lungall.gu.se; 2Department of Pulmonary Medicine, Koc University Hospital, 34010 Istanbul, Turkey; drbaranbalcan@gmail.com; 3Department of Molecular and Clinical Medicine/Cardiology, Institute of Medicine, Sahlgrenska Academy, University of Gothenburg, 40530 Gothenburg, Sweden; 4Department of Clinical Sciences, Respiratory Medicine and Allergology, School of Medicine, Lund University, 22185 Lund, Sweden; 5Division of Pulmonary, Allergy, and Critical Care Medicine, University of Pittsburgh School of Medicine, Pittsburgh, PA 15213, USA

**Keywords:** coronary artery disease, dyslipidaemia, obstructive sleep apnea, CPAP, randomized controlled trial

## Abstract

Dyslipidaemia is a well-known risk factor for coronary artery disease (CAD), and reducing lipid levels is essential for secondary prevention in management of these high-risk individuals. Dyslipidaemia is common also in patients with obstructive sleep apnea (OSA). Continuous positive airway pressure (CPAP) is the first line treatment of OSA. However, evidence of a possible lipid-lowering effect of CPAP in CAD patients with OSA is scarce. We addressed the effect of CPAP as an add-on treatment to lipid-lowering medication in a CAD cohort with concomitant OSA. This study was a secondary analysis of the RICCADSA trial (Trial Registry: ClinicalTrials.gov; No: NCT 00519597), that was conducted in Sweden between 2005 and 2013. In total, 244 revascularized CAD patients with nonsleepy OSA (apnea–hypopnea index ≥ 15/h, Epworth Sleepiness Scale score < 10) were randomly assigned to CPAP or no-CPAP. Circulating triglycerides (TG), total cholesterol (TC), high-density lipoprotein (HDL) and low-density lipoprotein (LDL) levels (all in mg/dL) were measured at baseline and 12 months after randomization. The desired TG levels were defined as circulating TG < 150 mg/dL, and LDL levels were targeted as <70 mg/dL according to the recent guidelines of the European Cardiology Society and the European Atherosclerosis Society. A total of 196 patients with available blood samples at baseline and 12-month follow-up were included (94 randomized to CPAP, 102 to no-CPAP). We found no significant between-group differences in circulating levels of TG, TC, HDL and LDL at baseline and after 12 months as well as in the amount of change from baseline. However, there was a significant decline regarding the proportion of patients with the desired TG levels from 87.2% to 77.2% in the CPAP group (*p* = 0.022), whereas there was an increase from 84.3% to 88.2% in the no-CPAP group (n.s.). The desired LDL levels remained low after 12 months in both groups (15.1% vs. 17.2% in CPAP group, and 20.8% vs. 18.8% in no-CPAP group; n.s.). In a multiple linear regression model, the increase in the TG levels was predicted by the increase in body-mass-index (β = 4.1; 95% confidence interval (1.0–7.1); *p* = 0.009) adjusted for age, sex and CPAP usage (hours/night). CPAP had no lipid-lowering effect in this revascularized cohort with OSA. An increase in body-mass-index predicted the increase in TG levels after 12 months, suggesting that lifestyle modifications should be given priority in adults with CAD and OSA, regardless of CPAP treatment.

## 1. Introduction

Dyslipidaemia, especially low-density lipoprotein (LDL)–cholesterol, is a well-known risk factor for coronary artery disease (CAD), and reducing lipid levels is essential for secondary prevention in management of these high-risk individuals [1]. According to the National Health and Nutrition Examination Survey, 11.7% of people between the ages 20–39, and 41.2% of adults between ages 40–64 have elevated levels of LDL [2]. 

Obstructive sleep apnea (OSA) is also an important public health problem in developed countries, affecting 9% and 24% of middle-aged women and men, respectively [3]. Based on the longitudinal Wisconsin Sleep Cohort study, published in 2013, the prevalence estimates are even higher (17% of women, and 34% of men) age 30–70 [4]. Moreover, dyslipidemia [5] and obesity [4] co-exist among adult OSA populations. Mechanisms, such as chronic intermittent hypoxia [6,7,8], sleep fragmentation [9,10] and sympathetic overactivation [11], have been suggested to contribute to dysregulation in lipid profiles among patients with OSA. 

OSA along with dyslipidemia has been independently associated with an increase in all-cause mortality, vascular heart disease and stroke, further creating an important demand for efficient treatment [12]. Continuous positive airway pressure (CPAP) is the first line treatment of OSA [13]. An observational study over 30 years has shown that OSA patients treated with CPAP for longer than 5 years were 5.6 times more likely to survive [14]. Notwithstanding, evidence of a possible lipid-lowering effect of CPAP in CAD patients with OSA is scarce. Previous meta-analyzes showed smaller reductions in circulating triglycerides (TG), total cholesterol (TC) and high-density lipoprotein (HDL)–cholesterol levels [15,16], but none of the included studies in those meta-analyzes were conducted in cardiac cohorts. A later study by Huang et al. [17] addressed the impact of CPAP in a small sample of 65 non-obese adults with newly diagnosed CAD, and reported no effect of CPAP on the circulating lipid levels. 

In the current study, we addressed the effect of CPAP for 12 months as an add-on treatment to lipid-lowering medication in a CAD cohort with concomitant OSA. We have also examined whether or not the desired TG and LDL levels according to the recent guidelines of the European Cardiology Society (ECS) and the European Atherosclerosis Society (EAS) were reached in this cohort following CPAP treatment [1]. 

## 2. Materials and Methods

### 2.1. Study Population

The present study is a primary analysis of one of the secondary outcomes of the **R**andomized **I**ntervention with **C**PAP in **C**oronary Artery **D**isease and obstructive **S**leep **A**pnea (RICCADSA) trial, which was conducted in Sweden between 2005 and 2013. The RICCADSA cohort has been previously described elsewhere [18]. In brief, adults with a history of percutaneous coronary intervention (PCI) or coronary artery by-pass grafting (CABG) within 6 months prior to study start were consecutively invited to participate. The CAD patients were classified as having OSA (apnea–hypopnea index [AHI] ≥ 15/h) and no-OSA (apnea–hypopnea index [AHI] < 5/h) based on the home sleep apnea testing (HSAT). As previously described in detail [18,19], the Embletta^®^ Portable Digital System device (Embla, Broomfield, CO, USA) was used for the HSAT recordings, which included a nasal pressure detector, two respiratory inductance plethysmography belts and pulse-oximetry for recording heart rate and oxyhemoglobin saturation (SpO2). Apnea was defined as at least 90% cessation of airflow, and the hypopnea definition was based on the guidelines from 1999 as at least a 50% reduction in nasal pressure amplitude and/or in thoraco–abdominal movement for at least 10 s [20]. For the current protocol, 196 CAD patients with nonsleepy OSA (AHI ≥ 15/h, Epworth Sleepiness Scale (ESS) score < 10) from the randomized controlled trial (RCT) arm were included (Figure 1). As previously described [19], the 1:1 random assignment of the main trial was scheduled with a block size of eight patients (four CPAP, four controls) stratified by gender and revascularization type (PCI/CABG).

### 2.2. Epworth Sleepiness Scale

The ESS is a self-rating questionnaire, which includes eight items estimating the risk of dozing-off under eight different conditions [21]. The subjects who scored less than 10 out of 24 points were categorized as nonsleepy. 

### 2.3. Comorbidities

As previously described [19], anthropometrics, smoking habits, medical history of the study population as well as medications, including the use of lipid-lowering agents were obtained from the medical records. Participants with as a body-mass-index (BMI) ≥ 30 kg/m^2^ were categorized as obese [22]. 

### 2.4. Circulating Lipid Levels

Blood samples were collected after an overnight fasting using ethylenediaminetetraacetic acid and serum tubes in the morning (07:00–08.00 am). Circulating total cholesterol (TC), high-density lipoprotein (HDL), low-density lipoprotein (LDL) as well as triglycerides (TG) levels (all in mg/dL) were measured at baseline and 12 months after randomization. The desired TG levels were defined as circulating TG < 150 mg/dL, and LDL levels were targeted as <70 mg/dL according to the recent guidelines of the ECS and the EAS [1].

### 2.5. Statistical Analysis

The study sample distribution of demographics and clinical characteristics at baseline was examined using the descriptive statistics. The Shapiro–Wilk test was used to test normality assumption of the current data for all variables. Continuous variables were reported as median values with boundaries of the interquartile ranges (IQR), and the categorical variables as numbers and percentages. Between-group differences were tested by the Mann–Whitney U test for the continuous variables. The chi-square test was used to compare the subgroups on the categorical variables. The within-group differences were tested by Wilcoxon Signed-Rank test for the continuous variables and McNemar’s test for the categorical variables. All statistical tests were two-sided and a *p*-value < 0.05 was considered significant. Statistical analyses were performed using SPSS^®^ 26.0 for Windows^®^ (SPSS Inc., Chicago, IL, USA).

## 3. Results

### 3.1. Baseline Characteristics of the Study Population

A total of 196 patients with available blood samples at baseline and 12-month follow-up, who were on lipid-lowering treatment were included (94 randomized to CPAP, 102 to no-CPAP). As shown in Table 1, demographic and clinical characteristics as well as HSAT and circulating lipid levels at baseline were similar in participants allocated to CPAP vs. no-CPAP. All patients were on statin-treatment at baseline as a part of the secondary cardiovascular prevention guidelines at the time of the study period.

### 3.2. Outcomes

All patients remained on the statin therapy at 1-year follow-up. As illustrated in Figure 2, there were no significant between-group as well as within-group differences regarding the circulating levels of the TG, TC, HDL and LDL levels at 12-month follow-up compared to baseline across the CPAP and no-CPAP groups in the intention-to-treat population.

As shown in Figure 3, the desired TG levels at baseline were similar between the groups. However, there was a significant between-group differences in proportion of patients with the desired TG-levels after 12 months (*p* = 0.048). The significant within-group difference was observed in the CPAP group (87.2% at baseline vs. 77.2% after 12 months; *p* = 0.022). The proportion of participants with the desired LDL levels remained low after 12 months in both groups (15.1% vs. 17.2% in CPAP group, and 20.8% vs. 18.8% in no-CPAP group, respectively) with no significant within- and between-group differences.

As illustrated in Figure 4, there was a linear association between the changes in TG levels and BMI at 1-year follow-up. In a multiple linear regression model, the increase in the TG levels was predicted by the increase in BMI (β = 4.1; 95% confidence interval (1.0–7.1); *p* = 0.009) adjusted for age, sex and CPAP usage (hours/night).

In the entire study population randomized to CPAP, the median CPAP usage was 3.8 h/night (IQR 0.0–6.1 h/night), and 47 out 94 (50%) were classified as CPAP adherent. The adherence was defined as at least 4 h/night, corresponding all nights during the 1-yr period. Post-hoc analyzes of the patients stratified by CPAP adherence did not change the main findings of the study (data not shown).

## 4. Discussion

The current study found no additional effect of CPAP therapy on the circulating levels of lipids after 12 months in CAD patients with OSA who were already on lipid-lowering medication. The desired LDL levels remained low in both CPAP and no-CPAP groups, and this was unrelated to OSA severity at baseline and/or CPAP adherence. Moreover, we observed a decrease in the proportion of patients with the desired TG levels, indicating a worsening. However, in a multiple linear regression model, the increase in BMI was the only significant predictor of the increase in TG levels independent of age, sex, AHI at baseline and CPAP adherence.

To the best of our knowledge, this study is the first RCT evaluating the effect of CPAP intervention as an add-on treatment to lipid-lowering medication in revascularized CAD patients with nonsleepy OSA. Previously, Huang et al. [17] addressed the impact of CPAP in a small sample of 65 non-obese adults with newly diagnosed CAD, who were on standardized lipid-lowering medication. The authors reported no effect of CPAP on the circulating lipid levels, which is in line with our results. We additionally showed no improvement regarding the proportion of patients with the desired TG and LDL levels after CPAP treatment. The decrease in the proportion of patients with the desired TG levels after CPAP treatment and its association with the increase in BMI is clinically relevant, and should be taken into consideration in the management of CAD patients with concomitant OSA.

The RCTs in non-cardiac cohorts with concomitant OSA [23,24,25,26,27,28] have also demonstrated no significant changes in circulating lipid levels except for the study by Phillips et al. [28], in which reductions in the total cholesterol and triglyceride levels have been demonstrated after 2 months of CPAP treatment. Moreover, a previous meta-analysis by Xu et al. [15] included six RCTs with 741 individuals and showed a significant but modest decrease in TC without any changes in the other parameters. Another meta-analysis Lin et al. [16] including 699 participants demonstrated smaller reductions in TG and HDL in addition to the decreased levels of TC following CPAP treatment. Thus, no impact has been shown on the circulating LDL levels, which is the most important target, especially in the secondary cardiovascular prevention models [1].

Chronic intermittent hypoxia is considered to be the main mechanism via systemic inflammation, oxidative distress, endothelial dysfunction and atherosclerosis in the previous literature [29]. It has also been suggested that increased sympathetic system activation may modulate the hormone-sensitive lipoprotein activity in the adipose tissue, which triggers lipolysis [30]. Moreover, Trammell et al. [31] showed in a mice model that sleep fragmentation is related with impaired lipid mechanism. Recently, Martinez-Ceron et al. [32] demonstrated a significant association between dyslipidemia and severe OSA, which might be due to sleep fragmentation and increased sympathetic activity. Although these proposed mechanisms constitute a relevant rationale to expect positive effects of the treatment of OSA by CPAP, our study does not show any significant impact on the circulating lipid levels as an add-on treatment to lipid-lowering medication. A slight decrease in the proportion of patients with the desired TG levels, explained by the increase in BMI at the 1-yr follow-up, as well as the remaining low proportion of patients with the desired LDL levels in spite of medication, strongly emphasizes that lifestyle modifications should be given priority in adults with CAD and OSA, regardless of CPAP treatment.

Lifestyle interventions on the severity of OSA as well as on the metabolic abnormalities in adults with OSA are still being studied, and have been showing promising results [33]. Moreover, although CPAP is recommended as the first therapeutic choice for patients with OSA, there is also data suggesting that pharyngoplasty with barbed sutures may improve the metabolic profile of patients with OSA [34,35].

Of note, our results apply to the CAD patients with the nonsleepy OSA phenotype. The reason for not including the patients with the “sleepy” phenotype in the current analysis was to evaluate the effect of CPAP on circulating levels of lipids in a more scientific manner, since randomization of individuals with the “sleepy phenotype” would not be ethical (risk for traffic and work accidents). Whether or not the response to CPAP treatment regarding the lipid profiles differs between OSA patients with the “sleepy” versus “nonsleepy” phenotype is indeed interesting, and will be further analyzed in the entire RICCADSA population.

The current study has three main limitations. First, the power estimate for the main RICCADSA trial was conducted for the primary outcomes (composite of major cardiovascular and cerebrovascular events) [19] and not for the secondary outcomes, and thus, the sample size may not be sufficient to validate the additional effect of CPAP on the lipid profiles. Second, our results are not generalizable to individuals with OSA in general population or other clinical cohorts. Third, categorization of patients as “nonsleepy” was based on an ESS score less than 10, which may not be precise in a CAD population. However, this questionnaire is a generally accepted tool used in clinical cohorts [21], and objective measurements, such as the Multiple Sleep Latency Test [36], are time consuming, expensive and not realistic to use in large-scale trials in cardiac populations.

## 5. Conclusions

CPAP had no lipid-lowering effect in this revascularized cohort with OSA. An increase in BMI predicted the increase in TG levels after 12 months, suggesting that lifestyle modifications should be given priority in adults with CAD and OSA, regardless of CPAP treatment.

## Figures and Tables

**Figure 1 jcm-11-00273-f001:**
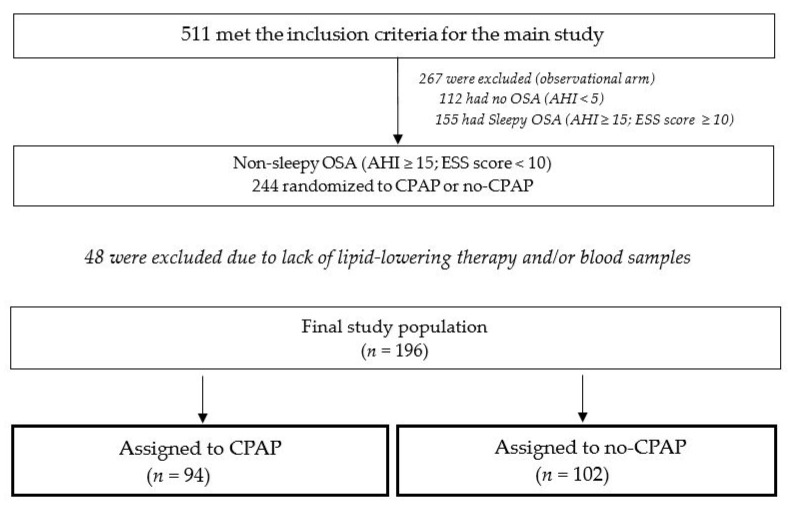
Consort flow chart of the study population.

**Figure 2 jcm-11-00273-f002:**
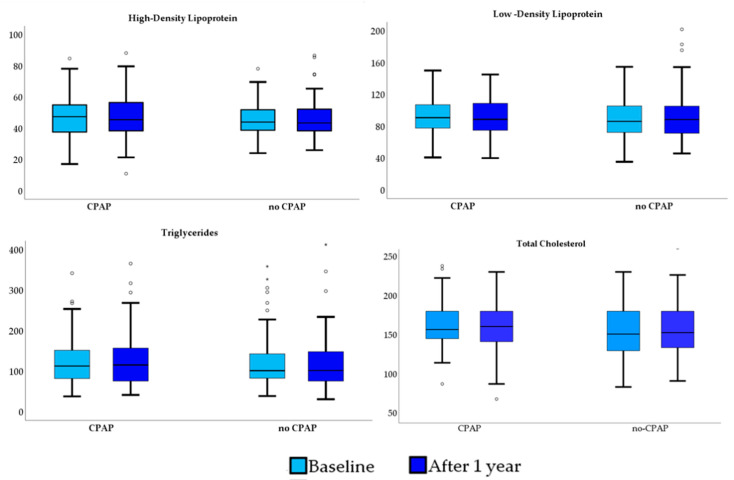
Distribution of the circulating lipid levels at baseline and after 12 months in patients in the CPAP and no-CPAP groups.

**Figure 3 jcm-11-00273-f003:**
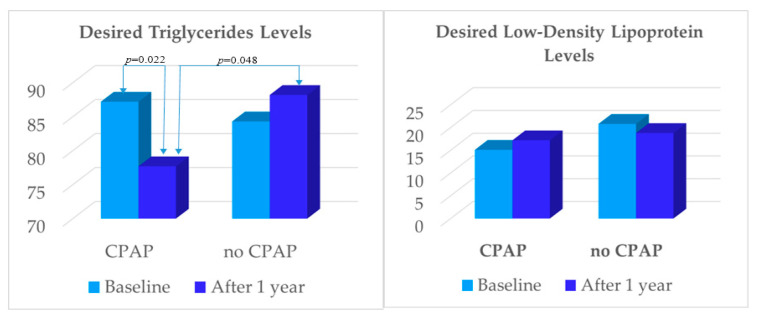
Proportion of participants with the desired triglyceride and low-density lipoprotein levels at baseline and after 12 months.

**Figure 4 jcm-11-00273-f004:**
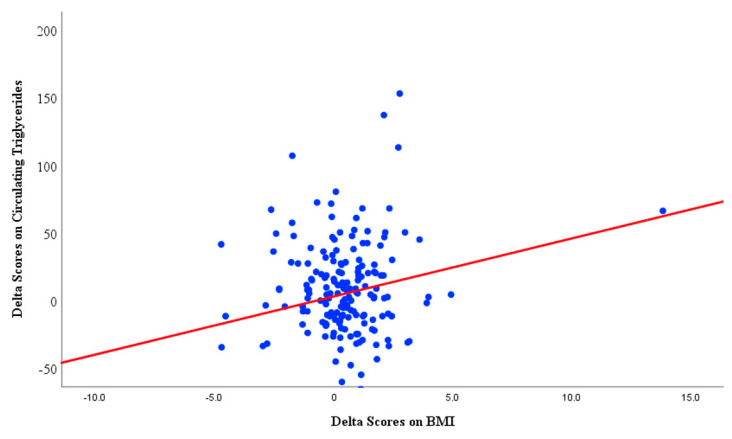
Multiple linear regression line for the association between change in body-mass-index and change in circulating triglyceride levels after 12 months.

**Table 1 jcm-11-00273-t001:** Baseline demographic and clinical characteristics of the CAD patients with nonsleepy OSA in the RCT arm.

	CPAP *n* = 94	No-CPAP *n* = 102	*p*
Age, yrs	65.4 (60.4–70.6)	67.3 (61.0–72.7)	0.243
Male,%	84.0	84.3	0.959
BMI, kg/m^2^	28.1 (25.7–30.2)	28.7 (26.6–30.6)	0.395
Obesity,%	27.7	29.4	0.786
PCI,%	76.6	76.5	0.984
AHI, events/h	23.5 (17.9–37.0)	24.9 (18.9–32.2)	0.648
ODI, events/h	13.5 (8.5–20.8)	12.6 (6.8–21.1)	0.502
ESS score	6 (4.0–8.0)	5.5 (4.0–7.0)	0.659
Total Cholesterol, mg/dL	158.6 (146.0–181.8)	152.8 (131.5–181.8)	0.107
HDL, mg/dL	47.4 (37.4–55.0)	43.9 (38.7–51.8)	0.290
LDL, mg/dL	90.1 (77.5–107.0)	87.0 (72.1–105.4)	0.309
Triglycerides, mg/dL	112.5 (81.3–151.7)	101.0 (81.4–143.0)	0.422
Glucose, mg/dL	101.0 (90.1–115.3)	97.3 (90.1–104.5)	0.166
Current smoker, %	17.0	14.7	0.657
AMI, %	57.4	48.0	0.188
Hypertension, %	66.0	59.8	0.373
Diabetes, %	26.6	18.6	0.182
Stroke, %	7.4	10.9	0.406
Lung Disease, %	4.3	9.8	0.132
Depression, %	4.3	3.0	0.619

Continuous data are presented as median and 25–75% quartiles. Categorical data are presented as percentage. Abbreviations: AHI, apnea hypopnea index; AMI, acute myocardial infarction; BMI, body mass index; CAD, coronary artery disease; CPAP, continuous positive airway pressure; ESS, Epworth Sleepiness Scale; HDL, high-density lipoprotein; LDL, low-density lipoprotein; OSA, obstructive sleep apnea; PCI, percutaneous coronary intervention; RCT, randomized controlled trial.

## Data Availability

Individual participant data that underlie the results reported in this article can be obtained by contacting the principal investigator of the RICCADSA trial; yuksel.peker@lungall.gu.se.

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
