# Peer review of "CPAP Intervention as an Add-On Treatment to Lipid-Lowering Medication in Coronary Artery Disease Patients with Obstructive Sleep Apnea in the RICCADSA Trial"

_jcm, 2022, doi:10.3390/jcm11010273_

Round 1

Reviewer 1 Report

Reviewer comments: Manuscript ID: jcm-1515317 Thank you for the opportunity to review the manuscript “CPAP Intervention as an Add-on Treatment to Lipid-lowering Medication in Coronary Artery Disease Patients with Obstructive Sleep Apnea in the RICCADSA Trial” written by Dr. Celik et al. This research article investigated the add-on lipid lowering effect of CPAP therapy to medication for patients with obstructive sleep apnea and coronary artery disease, using the secondary analysis design of the RICCADSA trial. The contents of this manuscript obviously include the important aspect of current clinical setting in all over the world. However, I think that several concerns (some of them seems serious) should be considered for publication as follows. Comments; 1. Page 2, study population, in Materials and Methods, Figure 1 Why 155 sleepy OSA patients were excluded in this analysis? I understand that the RISCCADSA randomized controlled trial (ref #31) was for non-sleepy OSA patients. However, I cannot find the scientific reason in the purpose of this study to distinguish sleepy OSA from non-sleepy OSA. 2. Page 2, home sleep apnea test (HSAT), in Materials and Methods The information of HSAT used as a diagnostic tool in this study should be more precisely described (i.e., which device, which company). 3. Page 4-5, change of cholesterol level, in Results The authors described in the Results (Figure 2) that there were no significant between-group as well as within-group difference regarding the circulating levels of the TG, TC, and LDL at 12-month follow-up compared to baseline across the CPAP and no-CPAP groups. If so, please demonstrate the more precise information of lipid-lowering medication in this study population (n= total 196) (i.e., which type, how many, whether the medication was changed during the study period or not). Otherwise, readers can misunderstand the results of the study (add-on effect of CPAP), I think. 4. Page 5, Figure 3 Please clearly show which parts (bars vs. bars) are statistically difference in the Figure. 5. Page 6, CPAP adherence, results Post-hoc analyzes of the patients with CPAP adherence (defined as at least 4 h/night) were conducted. I also think that this is very important. But more information is needed. How many patients in CPAP group (n=94) reached the good CPAP adherence (at least 4h/night)? Additionally, mean value of daily CPAP usage / 30 days (or certain time) should be also described. 6. (minor comments) Page 2, 2nd paragraph, Introduction A previous paper in 1993 (ref #3) were cited to explain the study background regarding the public health problem of OSA in developed country (i.e.; 9%-24% of middle-aged women and men can be OSA). However, since life styles are changing from 1993 to 2021, a little more recent information would be better (if any).

Reviewer 2 Report

  • Obstructive sleep apnea (OSA) is a very common condition often associated with central obesity. OSA contributes to the pathogenesis of metabolic abnormalities, including type 2 diabetes, metabolic syndrome and non-alcoholic fatty liver disease. New perspectives in the care of patients with OSA have been opened by promoting lifestyle interventions, such as diet and exercise programs that could improve both OSA and metabolic profile. Lifestyle interventions on nocturnal respiratory disorders and metabolism profile in patients with OSA are still being studied, showing promising results. please cite doi:10.1183/09059180.00003413
  • line 16, although CPAP is recommended as the first therapeutic approach for osas, the collapse of the upper airways due to instability of the pharyngeal lateral walls has shown significant pharyngoplasty results with barbed sutures, reducing the metabolic profile of patients. please cite doi:10.1007/s00405-016-4290-0 and doi:10.1016/j.amjoto.2021.103197.
  • line 24, add ''however the sample was not sufficient to validate such evidence''
  • which guidelines were used for the randomization? and for the study protocol?
  • use the consort flow chart
  • which PSG was performed? guidelines?
  • it would be better 50% randomization. Why not?
  • interesting figures
  • line 46, add a reference

Discussion

  • line 15, age was demonstrated as a predictive factor for OSA severity, with worse outcomes for elderly patients. please cite doi:10.20452/pamw.15283
  • recently a study reported that CPAP therapy led to decrease
    of 9.76 and 3.49 mmHg in systolic and diastolic blood pressures,
    respectively. Also, LDL decreased to 6.27mg/dl and HDL increased
    to 0.75 mg/dl (P<0.001) with treatment. The changes of other
    variables were not significant (P>0.05). please cite Asgari A, Soltaninejad F, Farajzadegan Z, Amra B. Effect of CPAP Therapy on Serum Lipids and Blood Pressure in Patients with Obstructive Sleep Apnea Syndrome. Tanaffos. 2019;18(2):126-132.

Round 2

Reviewer 1 Report

The manuscript was appropriately revised.